# Lichen Biodiversity and Near-Infrared Metabolomic Fingerprint as Diagnostic and Prognostic Complementary Tools for Biomonitoring: A Case Study in the Eastern Iberian Peninsula

**DOI:** 10.3390/jof9111064

**Published:** 2023-10-31

**Authors:** Patricia Moya, Salvador Chiva, Myriam Catalá, Alfonso Garmendia, Monica Casale, Jose Gomez, Tamara Pazos, Paolo Giordani, Vicent Calatayud, Eva Barreno

**Affiliations:** 1Instituto Cavanilles de Biodiversidad y Biología Evolutiva (ICBiBE)—Departament de Botànica, Universitat de València, C/Dr. Moliner, 50, Burjassot, E-46100 València, Spain; salvador.chiva@uv.es (S.C.); tamara.pazos@uv.es (T.P.); eva.barreno@uv.es (E.B.); 2Department of Life Sciences, University of Trieste, Via L. Giorgieri 10, 34127 Trieste, Italy; 3Instituto de Investigación de Cambio Global (IICG), Department of Biology and Geology, Physics and Inorganic Chemistry, School of Experimental Science & Technology, Rey Juan Carlos University, Av. Tulipán s/n, Móstoles, E-28933 Madrid, Spain; myriam.catala@urjc.es (M.C.); jose.gomezs@urjc.es (J.G.); 4Instituto Agroforestal Mediterráneo, Departamento de Ecosistemas Agroforestales, Universitat Politècnica de València, E-46022 València, Spain; algarsal@upvnet.upv.es; 5Department of Pharmacy, University of Genova, Viale Cembrano, 4, 16148 Genova, Italy; monica.casale@unige.it (M.C.); paolo.giordani@unige.it (P.G.); 6Fundación CEAM, Charles R. Darwin, 14, Paterna, E-46980 València, Spain; vicent@ceam.es

**Keywords:** air quality, Maestrazgo-Els Ports, epiphytic lichens, IAP (Index of Atmospheric Purity), DI (Damage Index)

## Abstract

In the 1990s, a sampling network for the biomonitoring of forests using epiphytic lichen diversity was established in the eastern Iberian Peninsula. This area registered air pollution impacts by winds from the Andorra thermal power plant, as well as from photo-oxidants and nitrogen depositions from local and long-distance transport. In 1997, an assessment of the state of lichen communities was carried out by calculating the Index of Atmospheric Purity. In addition, visible symptoms of morphological injury were recorded in nine macrolichens pre-selected by the speed of symptom evolution and their wide distribution in the territory. The thermal power plant has been closed and inactive since 2020. During 2022, almost 25 years later, seven stations of this previously established biomonitoring were revaluated. To compare the results obtained in 1997 and 2022, the same methodology was used, and data from air quality stations were included. We tested if, by integrating innovative methodologies (NIRS) into biomonitoring tools, it is possible to render an integrated response. The results displayed a general decrease in biodiversity in several of the sampling plots and a generalised increase in damage symptoms in the target lichen species studied in 1997, which seem to be the consequence of a multifactorial response.

## 1. Introduction

Lichens are one of the most emblematic examples of the mutualistic associations between one heterotrophic ascomycetous or basidiomycetous fungi (the mycobiont) and populations of photosynthetic green microalgae (phycobionts) and/or cyanobacteria (cyanobionts) (the photobionts) [1]. Aside from these major lichen symbionts that shape their unique symbiosis into a thallus, an indeterminate number of other microscopic organisms, including non-photosynthetic bacteria, accessory fungi, and microalgae, co-occur, which are intermingled in these associations [2,3,4,5]. Given all of these new “players”, the traditional lichen paradigm based on a mycocentric perspective is evolving into a broader concept of the lichen holobiomes [1,6,7]. Therefore, lichen thalli are considered microecosystems in which numerous different symbiotic partners can interact [8,9,10,11,12]. The fine functioning balance of thalli is one of the reasons for their recognition as bioindicators.

These organisms lack protective tissues, so they readily absorb water, nutritive substances, and gases directly from the atmosphere [13]. Due to these physiological peculiarities, lichens are sensitive to a suite of anthropogenic disturbances, such as atmospheric pollution, climate change, and forest management [14]. In fact, they have been widely used as biomonitors [15], both as bioaccumulators of trace elements [16,17,18,19] and as bioindicators [20,21,22,23]. Therefore, lichens are powerful and fast bioindicators of drivers of both small-scale and global change [24]. 

Specifically, epiphytic lichens can be used in different ways as air quality bioindicators as they act as early warning systems [23,25,26,27,28]. Biomonitoring methods based on the diversity of epiphytic lichens are among the most used worldwide [15]. In addition, evidence of pollutant impacts can be observed as symptoms of damage in lichen thalli, involving morphological and physiological alterations and fluctuations in lichen growth [29,30,31]. In this way, lichens can provide information related to pollutant concentrations in the monitored area [23,32,33]. Air pollutants limit the abundance of highly sensitive species and cause visible physiological changes to more tolerant species, such as damage to and discoloration of the thallus structure [29]. Pollutants, such as SO_2_ or heavy metals, cause changes in photobiont membrane permeability and increased electrolyte loss, especially K^+^ loss [34,35,36,37], and they can cause changes in thallus colour [23,38]. The degradation of surfaces renders the lichen more sensitive to dehydration, to variations in light intensity [27,39,40], and to invasion by pathogenic organisms [41,42] as a consequence of alterations to the cell membranes or death of the photobionts.

Both the species richness of epiphytic macrolichens and changes in the physical structures of specific species can be used to indicate air quality in an area [23,37]. Apart from assessing the effects of gaseous pollutants, lichen biomonitoring approaches were recently extended to a suite of other anthropogenic and ecological disturbances, such as land management practices and climate change [43,44]. Several aspects of lichen diversity (e.g., species richness and abundance, species composition, indicator species, functional traits, and groups) are usually considered in air quality monitoring, sustainable forestry, or ecosystem functioning [45,46,47].

In the Valencian Community (eastern Spain), the case of the Andorra Thermal Power Plant stands out because it highlighted the usefulness of lichens as biomonitors. This plant, located in Teruel province (Figure 1), became operational in 1981 and caused a significant environmental deterioration in the surrounding areas due to important emissions of pollutant gases. In 1988, twenty-five municipalities in the bordering province of Castellón (Valencian Community) filed a complaint for ecological crime. The complaint was dropped after an agreement that included a commitment to invest in the reduction of SO_2_ emissions from the power plant. In 1992, the commitments reached were materialised through the installation of filters for SO_2_ and the construction of a desulfurization plant [48]. In 1994, an environmental monitoring commission was created. A set of air quality monitoring stations to supervise pollutant emissions was established, which were located in the direction of prevailing winds NW–SE from the power plant. In 2018, and after reporting that the plant could not achieve the standards of the European environmental directive on pollutant emissions, the intention to close the plant was announced, and it finally closed in 2020.

During these years, biodiversity studies to analyse the effect of the plant on the nearby forests were promoted [49,50]. In 1991, after several prospections from 1986, a lichen biomonitoring network was established by our research group in Teruel and the bordering province of Castellón [38]. Twenty biomonitoring plots were located within the area of influence of the prevailing winds coming from the thermal power plant. In that moment, we assessed the diversity of lichen communities with a methodology proposed by Nimis et al. [43], later adopted by the Convention on Long-Range Transboundary Air Pollution (CLRTAP) and the European Union (ICP_Forests) [47], which included the Index of Atmospheric Purity (IAP), a widespread tool used for the assessment of air quality in forest areas to determine the quality of ecological conditions for lichens [51,52,53]. Between 1989 and 1991, visible damage was reported on the thalli of several lichens, as well as on the leaves of trees and shrubs in different areas of the Valencian Community [54,55]. For this reason, different species of foliose and fruticose lichens were selected to quantify the visible symptoms of damage observed in the field. Therefore, in addition to a modified version of IAP measurement designed by Nimis et al. [56], the quantification of visible damage symptoms in a Damage Index (DI) for epiphytic lichens at 20 locations was implemented. The objective was to establish a biological monitoring network in the same trees and stations where the inventories were carried out for the calculation of the IAP, to follow, during short periods of time, the evolution of most sensitive lichens. A new methodology for monitoring the potential impact of atmospheric pollutants on macrolichens was hence developed [38]. The result of the different IAP and DI data collection campaigns in this network was published by Barreno et al. [38], who concluded that this area registered different air pollution impacts by winds from the thermal power plant, as well as photo-oxidants and nitrogen depositions from local and long-distance transport [57,58]. This methodology was also applied with reliable results in other biomonitoring studies in Spain, such as those conducted in areas impacted by the La Robla Thermal Power Plant in León [59,60,61], in the Natural Park of La Font Roja [62,63], and in the Sierra de Espadán [64]. Although the power plant is currently closed and these types of studies are no longer conducted in the area, our biomonitoring network has been maintained. This allows us to perform temporal studies about the health of the forests in this area of the Iberian Peninsula. 

Despite the proven effectiveness and the diagnostic and prognostic potential of lichen biomonitoring, the important costs in terms of time and highly qualified human resources prevent its application in routine environmental monitoring plans [65]. Casale et al. [66] and Malaspina et al. [67] successfully proposed the use of near-infrared spectroscopy (NIRS) for generating a ‘fingerprint’ of lichens capable of discriminating between samples from polluted and non-polluted areas. Vibrational spectroscopy records the properties of matter by pulses of energy applied to a sample; the range extending from 750 nm to 2500 nm is known as near-infrared spectroscopy [68]. NIR spectra are complex and distinctive, constituting a global molecular fingerprint (GMF) [69] that allows for classifying specimens according to the collection site or microsite within the same population [70]. NIRS is a precise, fast, and low-cost technique that does not destroy or modify the properties of the sample, allowing many analyses to be performed in a very short period, making it an advantageous choice for routine biomonitoring. In addition, it is safe for the environment because the sample does not require any pretreatment or the use of chemical reagents, and it generates minimal waste, thereby meeting the green analytical revolution requirements. Although NIR spectroscopy has been little exploited for biomonitoring purposes, this study investigates its usefulness as a complementary tool to the IAP and DI for epiphytic lichens.

In this study, we reevaluate the lichen community diversity and structure and quantify the visible damage in nine epiphytic lichens in seven localities of this previously established biomonitoring network. We consider forest systems in the Valencian Community to examine the effect of environmental changes (climatic or pollution) on the diversity and morphological damage in lichen communities, and we demonstrate that lichens are suitable biomonitors to show temporal variation in the effects of anthropogenic impacts. We include the data from air quality stations to relate the differences observed in the lichen biomonitoring locations with environmental or climatic variations. Moreover, we test that by integrating complementary methodologies into biomonitoring tools, it is possible to render an integrated response from the point of view of biological organisation.

## 2. Materials and Methods

### 2.1. Biomonitoring Network and Air Quality Monitoring Stations

The selection of the network stations in 1991 was determined according to the presence of different phorophytes (i.e., tree species lichens grow on), potential vegetation, and bioclimatic conditions [71]. The stations were also selected according to the pH of the bark of the main phorophytes: acid (*Pinus*) and neutral–basic (*Quercus*). In this study, seven locations of a previously established biomonitoring network in Teruel and Castellón were reevaluated (Figure 1 and Appendix A). These plots were located within the area of influence of the prevailing winds coming from the thermal power plant except, for Toro-*Pinus* and Toro-*Quercus,* considered control plots, which were 120 km away from the power plant. 

Data regarding SO_2_, O_3_, and temperature from seven air quality monitoring stations from the Valencian Community Air Quality Network from 1996 to 2023 were included: Corachar, Morella, Vilafranca, Cirat, Viver, Villar Arzobispo, and Torrebaja (Figure 1). These stations were selected due to their proximity to the biomonitoring network locations being reevaluated.

### 2.2. Index of Atmospheric Purity

The selected trees examined in the 1997 campaign were located within a radius of 0.5 km around the station. These trees were marked using visible metal plaques, and they met the following characteristics: (a) trunk diameter between 20 and 40 cm, (b) trunk inclination lower than 20°, (c) at least 3 km away from road networks, (d) not included in excessively closed tree formations, and (e) no regrowth at the base. In 2022, sampling for IAP was carried out on the same trees analysed during the 1997 campaign. In case it was not possible to locate them, other trees with the same characteristics were selected.

The composition and structure of the lichen community were assessed with a modified version of the Index of Atmospheric Purity (IAP) [72] designed by Nimis et al. [56]. To compare the results obtained in 1997 and 2022, we have now used the same methodology. This index evaluates the frequency of occurrence of different epiphytic lichen species over an area subdivided into 10 squares. The area is subdivided by a quadrant of 50 cm in length and 20 cm of width, and it was always located in the area of greatest lichen coverage [43,73,74]. To compare the results, the quadrant dimensions were identical, 20 × 50 cm, subdivided into 10 squares of 10 × 10 cm, and the quadrants were placed, whenever possible, in the same orientation as in the 1997 campaign (Appendix A).

The epiphytic lichen diversity index was calculated for each tree (IAP tree) and for the station (IAP station), according to the following expressions:IAP tree = Σf_i,_(1)
IAP station = (ΣIAP tree)/m(2)
where f = frequency of occurrence of each species within each grid for each tree i (taking values between 1 and 10), and m = number of trees per station.

For all of the species registered in the IAP inventories, we retrieved the ecological indicators proposed by Nimis [75], and we assigned for each species the value of the maximum category that can be achieved from ITALIC 7.0—the information system on Italian Lichens [75] on selected traits (Appendix A). For the growth form, we recorded the categories crustose, foliose broad-lobed (*Parmelia*-type), foliose narrow-lobed (*Physcia*-type), and fruticose, and for reproductive structures, we recorded apotecia, isidia, and soredia.

### 2.3. Lichen Damage Index (DI)

For the quantification of visible damage, we decided to use macrolichens as bioindicators since Nascimbene et al. [76] demonstrated that a sample of macrolichens, or even only large-lobed foliose lichens, allows the diversity to be reliably estimated. Moreover, in this type of lichen, it is easier to visualise a gradient of damage and, therefore, to obtain a more robust assessment. The selection of lichen species was based on the results of damage and/or chlorophyll activity evaluation obtained in the previous campaigns [32,38] and, due to their wide distribution in the territory, according to the phorophyte that may occur, because the pH of the bark determines the presence of lichen species. The nine species of lichens selected are the foliose growth form lichens: *Pleurosticta acetabulum* (Neck.) Elix and Lumbsch, *Parmelina carporrhizans* (Taylor) Poelt and Vězda, *Parmelia serrana* A. Crespo, M.C. Molina, and D. Hawksw., *Xanthoria parietina* (L.) Th. Fr., *Pseudevernia furfuracea* (L.) Zopf. and *Physcia aipolia* (Humb.) Fürnr. (narrow laciniea). In the case of the fruticose growth form, the following were selected: *Anaptychia ciliaris* (L.) A. Massal., *Ramalina fraxinea* (L.) Ach, and *Ramalina farinacea* (L.) Ach.

Previously, a methodology was developed to monitor the impact of atmospheric pollutants on macrolichens, which was inspired by publications by Sigal and Nash [77] and Nash and Sigal [78] in which they analyse descriptively the variation in morphological patterns due to the impact of photo-oxidants on different species in southern California. In addition, the methodological guidelines proposed for the quantification of damage to conifer needles by the USDA Forest Service, which facilitate the annual monitoring of visible symptoms, were considered [79]. Thus, Barreno et al. [38] developed a semi-quantitative evaluation protocol for the symptoms of visible damage (DI). On each tree, 10 thalli from each of the nine selected species were measured whenever possible. The visible symptoms (colour changes, necrosis, twisting, etc.), their location (centre of thallus or lobes), and the percentage of affected surface of each specimen were recorded in the field. In order to reduce the subjectivity of the evaluations, we tried to harmonise the symptoms recorded (see the template used in the field in Appendix A). To calculate the Damage Indexes (DIs) for each target species, the data taken in the field had to be classified according to the number of thalli, their size, and the degree of damage symptoms assigned, with ranges that vary from 1 to 5, with 1 being no visible damage and 5 being necrosis in 90% of the thallus/dead thallus (see Appendix A for an example of ranges of DI for selected species). To compare the results obtained in 1997 and 2022, we used the same methodology and recorded the DI in the same trees whenever possible.

To assign the damage category (1–5), in the case of a thallus presenting multiple damages, the maximum DI value of the symptoms present in the lichen was assigned: max (DI). A total of 2388 thalli were considered: 956 in 1997 and 1432 in 2022.

### 2.4. Statistical Analyses of IAP and DI

Original data and scripts for the analysis can be found in the Appendix A.

Statistical analyses, tables, and figures were made using R language (R Core Team 2023) with RStudio gui (RStudio Team 2023) [80,81]. To manage data tables’ structures and tables’ dplyr [82] and tidyr, packages [83] were used. The packages used for the graphics were: ggplot2 [84], ggpmisc [85], ggstatsplot [86], BiodiversityR [87], ggrepel [88], and RColorBrewer [89]. Also, the readxl [90] package was used to read the data from Excel files. And, the function kable from the knitr package [91] was used to render the tables.

The change in the Index of Atmospheric Purity (IAP) and the Damage Index (DI) between the years 1997 and 2022 was analysed first using the Student’s T test to compare between years for each locality. The change in species frequency was represented for each species and locality using a heatmap. A representation of the overall change was made comparing the Non Metric Dimensional Scaling results from both years. The effect of the species characteristics [75] on the lichens abundance (IAP) change and Damage Index (DI) was analysed using linear models for the numeric characteristics and ANOVA tests for reproductive structures and growth forms.

### 2.5. Acquisition of Visible–Near Infrared Spectra

The visible–near infrared (Vis-NIR) spectra of milligram samples of a subset of 195 thalli were acquired (Appendix A), including 26 from El Toro—*Pinus*, 50 from El Toro—*Quercus*, 28 from Cinctorres, 15 from Corachar, 27 from Bojar, 35 from Collado Gavilan, and 10 from Villarroya Pinares. Of these, 75 specimens presented a healthy appearance according to the visual Damage Index described above, whereas the rest showed some kind of damage. The subsets of thalli showing stains (64), parasites (9), or excessive reproductive structures (32) were also used for independent NIRS analyses.

ASD LabSpec^®^ 4 Standard-Res laboratory analyser (Malvern Panalytical, Malvern, UK) and the Indico Pro (version 6.5.6.1) were used. Samples frozen at −80 °C were thawed at room temperature (20–25 °C) in batches of 10. Each sample was placed on a calibrated white panel of spectralon. Measurements were performed directly on the surface of the lichen with a reflectance probe. Vis-NIR spectra were taken between 350 and 2500 nm at a resolution of 1 nm average of 32 scans. The calibrated white reference panel of spectralon was also used as a reference.

### 2.6. Data Analysis of FT-NIRS Spectra

The aquaphotomics approach, based on the study of the interaction between water and electromagnetic radiation, was chosen; for this reason, the region of the NIR spectrum corresponding to the first overtone of the OH stretching vibrations (1300–1600 nm) was selected and used in this study.

NIR spectra were organised in data matrices with as many rows as samples (195) and as many columns as variables (301). The categorical variables (or additional information) were the following five: reproductive structures (as for DI), growth form, reproduction strategy, phorophyte, and bioclimatic belt. Data matrices were split into training and test sets. The test set always included 58 samples chosen randomly.

The first step in data evaluation was pre-processing: for raw data pre-processing, a multiplicative scatter correction (MSC) was applied, according to the aquaphotomics procedure, allowing us to properly address the absorption bands without considering global intensity effects. Then, the data were mean centred. The pre-treated data were submitted to an exploratory processing step (Principal Component Analysis—PCA) and then to classification analysis (PLS-DA).

PCA was performed as a multivariate display method on the NIR spectra of thalli to extract the useful information embodied within the data and to visualise the data structure, after multiplicative scatter correction (MSC) and column centring as data pretreatments.

PLS-DA is a classification method and a variant of partial least squares regression (PLS-R) that is used when the response variable is categorical; mathematical operations of PLS regression and PLS-DA are nominally the same, with the major difference being the response that is predicted. Instead of containing continuous sample responses (like in PLS), the Y matrix in PLS-DA contains sample membership in categorical responses (e.g., 0, 1 for two classes).

PLS-DA is a dimensionality reduction technique, and it can be seen as a compromise between the usual discriminant analysis and a discriminant analysis on the principal components of the predictor variables (X). A dimensionality-reducing transformation results in latent variables (LVs), which are linear combinations of the original spectral variables that attempt to explain the maximum covariance between X and Y.

External validation was performed using the NIR spectra of 58 samples assigned to the test sets and a number of latent variables previously determined through cross-validation. The model performance was evaluated by sensitivity, specificity, and classification and prediction errors.

## 3. Results

### 3.1. IAP

The epiphytic lichen community diversity and structure were compared in seven locations between 1997 and 2022. Most of them showed a significant decrease in the IAP values; Villarroya Pinares maintained similar values in both campaigns, and Corachar showed a tendency to increase (Appendix A and Figure 2). The significant decrease in the values of the stations considered as controls in 1997 (Toro) with respect to 2022 is remarkable. In fact, the order of the stations in 1997 from the lowest to the highest IAP value is: Corachar, Villarroya Pinares, Bojar, Collado Gavilan, Toro-*Pinus*, Cinctorres, and Toro-*Quercus*. However, in 2022, it is: Corachar, Toro-*Pinus*, Bojar, Villarroya Pinares, Cinctorres, Collado Gavilan, and Toro-*Quercus*.

Change in species frequency was represented for each species and locality from 1997 to 2022 using a heatmap (Appendix A). Several species from the genera *Physcia*, *Hyperphyscia*, *Melanelixia*, and *Melanohalea* displayed a decrease in their abundance that was more remarkable in *Quercus* forests. In contrast, *Pseudevernia furfuracea* increased its abundance in the community, mainly in *Pinus* forests (Appendix A).

Figure 3 depicts the distances in species composition (NMDS) between trees from different locations in different years. Changes in the community structure can be noticed since 1997; in that year, the communities were more similar to each other, except for Villarroya Pinares and Corachar. In 2022, a separation of the communities according to the phorophyte was observed, and, in the case of Corachar, two different lichen communities were detected (see Figure 3).

Linear models for the numeric characteristics were performed to correlate the species change in abundance between the two campaigns with the ecological indicators. In this case, only the pH of the substratum and the tolerance to eutrophication (Figure 4 and Appendix A) were highly correlated (Pearson r = 0.74475008), and they can be related to the differences in the lichen community.

### 3.2. DI

Lichen morphological injury in the target macrolichens was compared between localities and years. The degree of damage symptoms assigned ranged from one to five. In this analysis, five of the locations showed a significant increase in the DI, except for Bojar, which maintained similar values in both campaigns, and Corachar, which showed a significant improvement in 2022 (Appendix A and Figure 5).

The change in DI between 1997 and 2022 for each species and locality was also depicted using a heatmap (Figure 6). *Pleurosticta acetabulum* and *Parmelia serrana* displayed significantly less damage in the lichen morphology.

Although not all species showed the same trend, in general, no effect in relation to the size was observed. Individuals with a larger size do not appear more damaged or with a higher DI (see Appendix A).

### 3.3. Air Quality Stations

For each station, the mean SO_2_ and O_3_ concentrations and temperature during the period between 1997 and 2022 are visualised in Figure 7. SO_2_ values decreased significantly along the years. Since 2016, SO_2_ the levels have been reduced to 3 µg/m^3^ in all the stations. Corachar, which received a much greater impact of SO_2_ during the first 12 years of measurement, showed a significant decrease since 2008.

The ozone values indicated that each locality has not experienced major fluctuations over the years, but, in general, values in Corachar and Morella were higher than in the rest of the localities (values of 80–90 vs. 50–70).

As regards the temperature, Toro and Collado Gavilan temperatures are traditionally higher, and Morella and Corachar are colder. Figure 7C shows that temperatures are becoming unified, and the differences between localities are lower.

### 3.4. Near-Infrared Aquaphotomics Analysis

We selected the parameters referring to the presence stains, excessive reproductive structures, and parasites in order to perform spectral analysis because of a closer link to the metabolome. PCA of thalli NIR spectra in the water band showed interesting results regarding the excessive reproductive structures parameter of damage (Figure 8).

Given that population studies rendered interesting data connecting biodiversity changes and some biological traits, we also performed some analyses to learn whether these traits were recognisable through the NIR molecular fingerprint. Interesting patterns were observed for lichen growth forms, reproduction strategy, phorophyte, and bioclimatic belt (Figure 9).

PLS-DA is a dimensionality reduction technique, and it can be seen as a compromise between the usual discriminant analysis and a discriminant analysis on the principal components of the predictor variables (X); a dimensionality-reducing transformation results in latent variables (LVs), which are linear combinations of the original spectral variables that attempt to explain the maximum covariance between X and Y. The results displayed in Table 1 show very good sensitivity and specificity of PLS-DA models for growth form and bioclimatic belt prediction using the thalli NIR fingerprint in the band of water (1300–1600 nm). Satisfactory results are also obtained for the prediction of the reproduction strategy and phorophyte, while the prediction of the appearance of excessive reproductive structures as a parameter of damage render moderately good sensitivity and specificity, which could probably be incremented with a larger number of individuals.

## 4. Discussion

Biomonitoring provides a direct measure of the state of the ecosystems’ health and the environmental quality of a territory [23,43]. In fact, biomonitoring techniques are nowadays widely applied for the establishment of conservation strategies for threatened species or habitats, as well as for their management [92]. Current environmental analyses using lichens provide high-resolution spatial tools for mapping the effects of environmental changes, at a low cost, compared to traditional pollution studies or climate monitoring stations [27,93,94,95].

In this study, we considered forest systems in the Valencian Community to examine the effect of environmental changes (climatic or pollution) on the diversity and morphological damage in lichen communities. Moreover, we demonstrated that lichens are suitable biomonitors to notice temporal variation in the effects of anthropogenic impacts. The results displayed a general decrease in diversity in several of the sampling stations and a generalised increase in damage symptoms in the target lichen species proposed in 1997, except in Corachar, which seem to be the consequence of a multifactorial response outlined below.

Biomonitoring methods based on the study of epiphytic lichen communities represent excellent tools for monitoring the effects of air pollutants over time, especially sulfur and nitrogen compounds and atmospheric particulate matter [43,46]. Some authors [29,96] have proven the correlation between modelled air pollution and IAP values based on epiphytic communities and also that IAP can reveal the locations of ecological degradation, which may be associated with low-level air pollution. Meanwhile, other authors suggest carefully interpreting lichen diversity data in terms of the direct effects of pollution in forest areas [15].

Here, the diversity index utilised allowed us to analyse the current state of the lichen community and its evolution over time. Most of the locations showed a significant decrease in the IAP values, but one showed a tendency to increase (Corachar). Our analyses indicated that variation in lichen communities seems to be related to eutrophication levels. Functional traits are considered suitable indicators for monitoring forest pollution and climate change [97,98,99]. The concurrent study of the functional diversity and the ecological optima of species, rather than considering shifts in species composition alone, can help in better monitoring directions in lichen biota changes over time [99]. Lichen functional groups for eutrophication and, more specifically, nitrogen tolerance have been extensively used to assess the critical level and critical load of nitrogen compounds in several forest ecosystems all over the world [100,101,102,103,104]. The impact of and increase in nitrogen deposition in recent years in our study area (Appendix A) is a key factor that should not be overlooked in biomonitoring studies.

In addition to the diversity indexes, the methodology developed included a study on damage symptoms in nine target lichen species. Most of these species are listed as occurring in moderately disturbed areas. The study of single indicator species, such as air-pollution-sensitive or more tolerant lichens, can provide useful information on air pollution and climate change impacts on forest sites. This approach builds on the assumption that air pollutants can cause physical and physiological alterations to lichens, resulting in visible injuries on their thalli (e.g., damage or discoloration) [25]. For example, in the case of *Parmotrema tinctorum* and *Usnea barbata*, the lichen morphoanatomy and photochemistry have been demonstrated to be efficient biomarkers for the effects of exposure to increasing concentrations of Cd [31].

In this context, the Finnish epiphytic macrolichen survey method proposes to monitor air quality by focusing on the presence of a set of macrolichen species and the abundance of and degree of damage suffered by two species: the highly sensitive *Bryoria* spp. and the more tolerant *Hypogymina physodes* [105]. A five-class scale of damage is used to assess to the most damaged individuals based on their appearance, the presence of spots, and colour variation. In the present study, the morphological analysis in the field is more detailed and aims to refine this type of methodology.

Mayer et al. [29], when adopting the Finnish method in southern Finland and northwestern Russia, suggested that the responses of single species may not be reliable bioindicators at low air pollution concentrations, but the authors themselves admitted that the selection of the species was inadequate. In the present study, almost all locations showed a significant increase in the DI, except for Corachar, which showed a significant improvement in 2022 with a healthier lichen community. Moreover, no effect between size and DI was detected: individuals with a larger size do not appear more damaged or with a higher DI. In our case, these results together with those of IAP are very consistent and reflect the robustness of our methodology for the sampled areas and the accuracies of the selected species.

The studies conducted from 1994 to 1997 and described in the aforementioned ENDESA report of 1998 [38] already showed a differentiation in the Corachar station. During those years, it presented the lowest values of IAP and species diversity and evolved negatively. These results were directly related to the incidence of the plume of the power plant. However, the current trend at this location is significantly different, as evidenced in our study, and this is linked to a decrease in the impact of SO_2_ in the area due to the inactivity of the power plant.

The proximity to the thermal power plant is a determining factor in the results obtained for Corachar, as had already been seen for other power plants [59,60,61]. Currently, this is no longer a determining factor for the deterioration of the localities because the authors of the report have already mentioned the existence of a multifactorial response. Our biomonitoring network is located in an area considered critical for the formation of photooxidants at the European level [106]. As in many territories, in the Iberian Peninsula, during the summer, the sea temperature is lower than that of the surrounding land, producing sea and hillside breezes. In general, the penetration of these breezes depends on the orography, but they can reach up to 100 km inland [58]. These breezes or winds with pollutants are driven by river basins.

In the case of the interior north of Castellón, there are large, uninhabited areas without industry, which a priori should be free of pollutants. However, the Mediterranean winds laden with anthropogenic substances from coastal industry are driven inland by the Mijares river basin, and this could explain the progressive deterioration detected in all biomonitored localities. Many of these substances are transported over long distances from their emission sources, and they mainly spread through wet and dry deposition. Indeed, forests may be affected by different types of atmospheric deposition, which can compromise their health and inner balance [107]. The problem caused by the dynamics of breezes mixed with pollutants from the coastal areas towards the inland territories of the areas here studied continues today and has even been intensified.

In addition, changes in the use of these territories should be considered, while not excluding the effects of climate change [28,95]. The observed homogenisation and increase in temperatures together with the change in agricultural use could be key to the decline observed in the former control locations.

Many aspects related to the interpretation of the lichen diversity data have not yet encountered a standard agreement between researchers, so new tools are currently developing in this field. The lichen Damage Index (DI) constitutes a valuable diagnostic tool for biomonitoring, but it requires highly qualified personnel with specific training to reduce the subjectivity during protocol application. Anatomical symptoms observed in thalli are the consequence of cellular events which, in the ultimate instance, originate in biochemical and metabolic alterations caused by pollutants and environmental stress. Vibrational spectra are objective and extremely fast measurements feasible for medium-qualified technicians, and the development of automated models correlating NIR spectra with observed DI could be a useful tool for biomonitoring. The aquaphotomics approach that uses the band of water between 1300 and 1600 nm is especially promising.

Here, we tested if by integrating NIR as a complementary methodology into biomonitoring tools it is possible to have an integrated response from the point of view of biological organisation. NIR aquaphotomics is an “omics” discipline established by Roumiana Tsenkova and thoroughly presented in a recent review [108]. The main subject of this approach is to understand the integrative role of water in biological and aqueous systems by monitoring how the water spectrum changes under various perturbations. In contrast to the traditional NIR spectroscopy studies, in which the water absorption band is considered to be masking the real information, aquaphotomics considers the water spectral pattern as the main source of information [109,110]. The implementation of new techniques allowing fast and low-cost biomonitoring is an urgent need in the present scenario of global change characterised by climate warming and water scarcity. Near-infrared allows the acquisition of fingerprints characteristic of the metabolic state of an organism, including the water structure. To have a proof of concept that NIR aquaphotomics may facilitate and speed up the study of biological traits as well as biomarkers of damage for biomonitoring, we acquired the spectra of a subset in which the samples’ index of visual damage was known.

No correlation between lichen DI and NIR spectra was found, probably due to the nature of some of the symptoms used, which do not relate directly to the metabolome (i.e., twisting, loss of surface). The PLS-DA model indicated a moderate correlation for excessive reproductive structures, suggesting that this compensatory response alters the molecular structure of water. Spectral characteristics indicate that the minute amounts of biomass used, in the milligram range, could constitute a limiting factor in our experimental design, which should be explored in the future. Larger biomass will enable the detection of minor peaks corresponding to relevant functional molecular groups.

Conversely, PLS-DA provided very good predictive models regarding biological traits, such as growth form and bioclimatic belt, while they are moderately good for reproduction strategies and phorophyte. As discussed above, functional traits are considered an improved tool for monitoring global change drivers, such as pollution and climate warming [97,98,99]. Our results demonstrate that the development of advanced and artificial intelligence predictive models using NIRS in the water spectral range will allow, for example, the detection of the evolution of the community towards a warmer or drier bioclimatic belt without taxon identification. Since Because the acquisition of NIRS spectra is technically very simple and fast and feasible for medium-qualified personnel, environmental biomonitoring costs could be dramatically reduced, and the frequency or extension could be increased. To this end, larger and more diverse lichen samplings should be implemented in the future to build robust and reliable models. Moreover, a detailed study of the spectral profile of water molecules, known as aquaphotome [108], could help unveil which are the specific water species associated with biological traits and the identification of water scarcity adaptation mechanisms and biomarkers, as recently suggested by Bruñas et al. [70].

Our results are a pioneer proof of concept of the usefulness of the near-infrared molecular fingerprint together with advanced statistical methods for a lichen community biological traits assessment as drivers of global change and climate warming.

## Figures and Tables

**Figure 1 jof-09-01064-f001:**
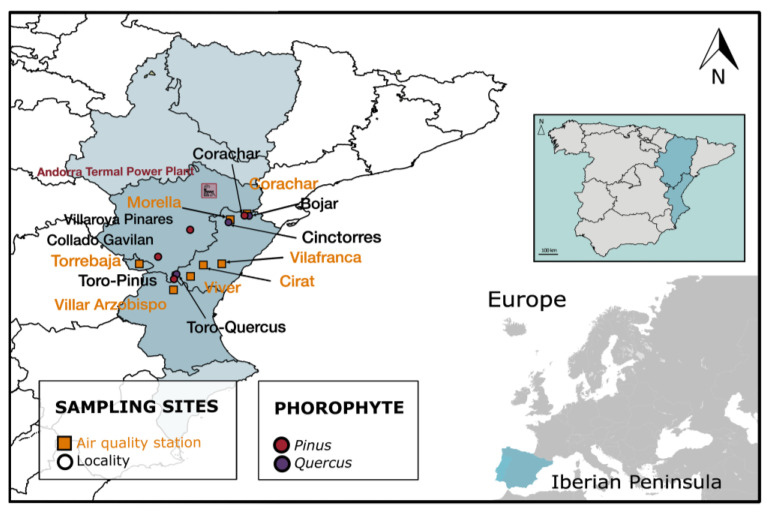
Map showing the seven stations of the biomonitoring network with circles. The colour of the circle indicates the phorophyte: orange—*Pinus* and purple—*Quercus*. Seven air quality monitoring stations from the Valencian Community Air Quality Network were included and written in a dark blue colour. The Andorra Thermal Power Plant is indicated and highlighted in red.

**Figure 2 jof-09-01064-f002:**
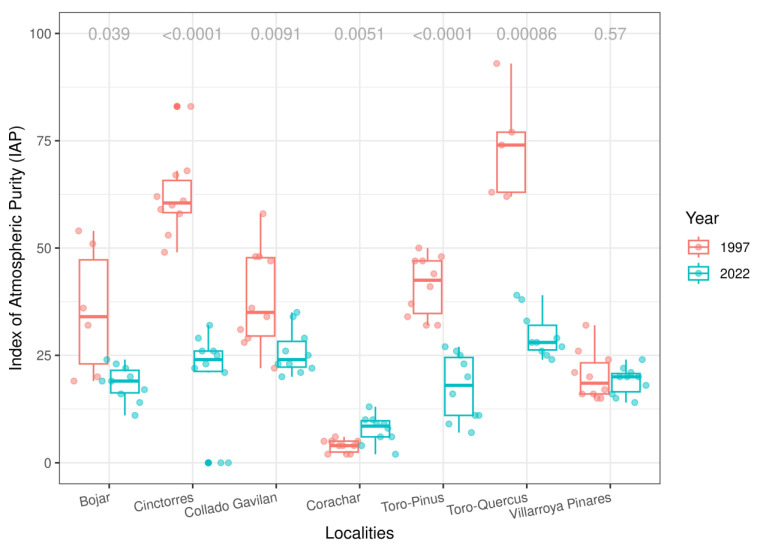
Index of Atmospheric Purity (IAP) comparison between years at each locality. *p* values of Student’s T test between years are at the top. Jittered dots represent individual IAP values for each tree.

**Figure 3 jof-09-01064-f003:**
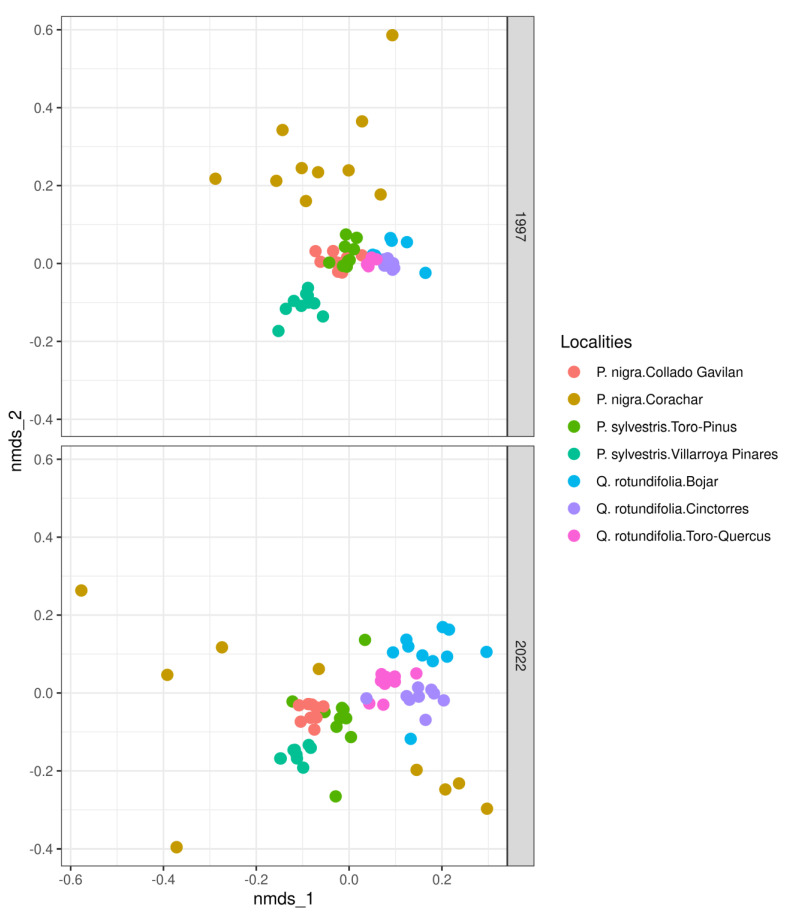
Distances of lichen species composition between sampled trees for the different localities. Type of phorophyte is indicated together with the name of each locality.

**Figure 4 jof-09-01064-f004:**
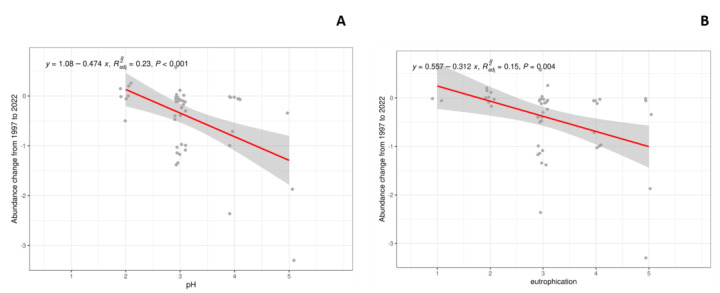
Effect of species’ characteristic pH (**A**) and eutrophication (**B**) in the abundance change between 1997 and 2022.

**Figure 5 jof-09-01064-f005:**
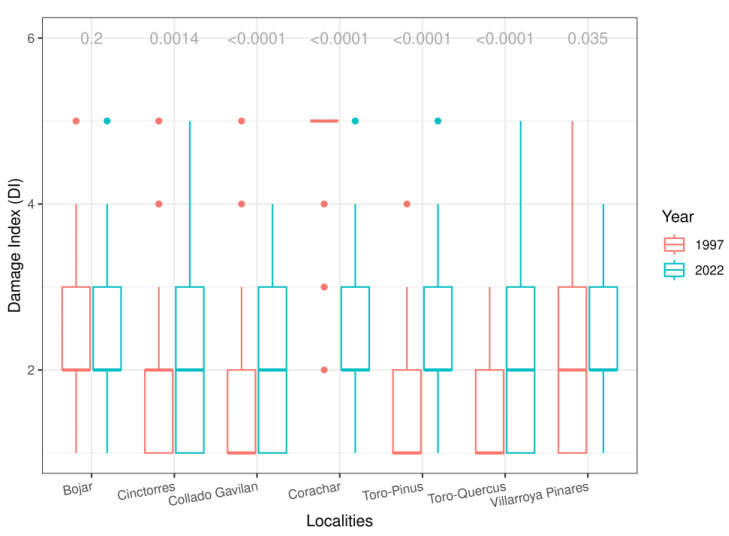
Damage Index (DI) values between localities and years. *p* values of Student’s T test between years at the top.

**Figure 6 jof-09-01064-f006:**
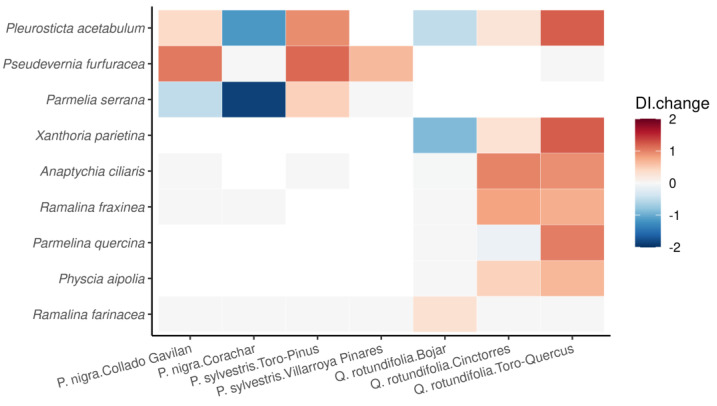
Mean change in the Damage Index (DI) between 1997 and 2022 for each species in each locality. Blue colour indicates a decrease in DI while red colour indicates an increase in DI. Species are ordered from larger to smaller absolute change.

**Figure 7 jof-09-01064-f007:**
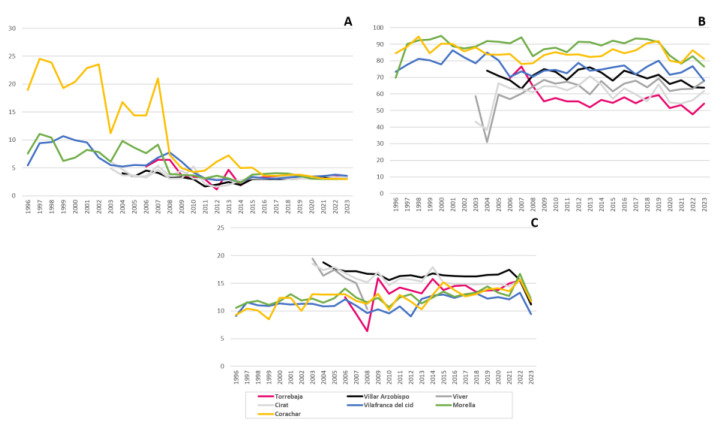
Annual mean values for SO_2_ µg/m^3^ (**A**), O_3_ µg/m^3^ (**B**) and temperature (**C**) during the period between 1997 and 2022 for seven air quality stations (indicated in different colours) from the Valencian Community Air Quality Network.

**Figure 8 jof-09-01064-f008:**
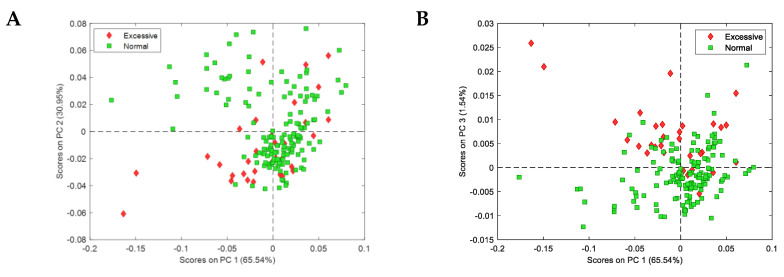
Principal component analysis of the NIR spectra of a subset of 195 thalli according to the appearance of excessive reproductive structures as a parameter of visible damage. Samples are represented using different colours according to normal (green square) or excessive (red diamond) reproductive structures seen. Score plots in the space PC1–PC2 (**A**) and PC1–PC3 (**B**).

**Figure 9 jof-09-01064-f009:**
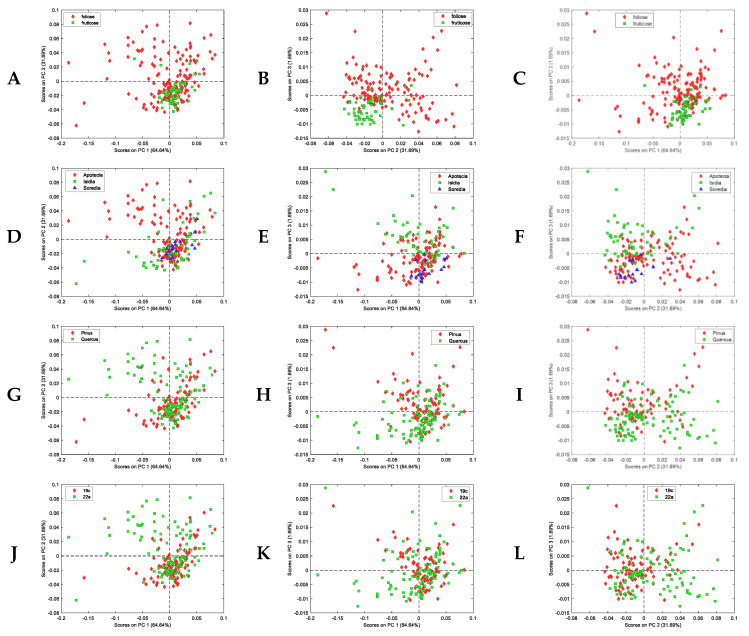
Principal component analysis (PCA) of the NIR spectra of a subset of 195 thalli according to different biological traits. Score plots in the space of: fruticose thalli (green square) and foliose (red diamond) according to growth form PC1-PC2 (**A**), PC2-PC3 (**B**), and PC1-PC3 (**C**); species having apotecia (red diamond), isidia (green square), and soredia (blue triangle) as reproductive structures PC1-PC2 (**D**), PC1-PC3 (**E**), and PC2-PC3 (**F**); Pinus sp. (red diamond) and Quercus sp. (green square) as phorophyte PC1-PC2 (**G**), PC1-PC3 (**H**), and PC2-PC3 (**I**); species associated with 19c supramediterranean (red diamond) and 22a subhumid (green square) bioclimatic belts PC1-PC2 (**J**), PC1-PC3 (**K**), and PC2-PC3 (**L**).

**Table 1 jof-09-01064-t001:** PLS-DA results. These figures of merit were calculated through cross-validation (CV) and prediction based on the test set (pred).

PLSDA	CLASS	LVs	Sn (CV)	Sp (CV)	Er (CV)	Sn (Pred)	Sp (Pred)	Er (Pred)
Reproductive structures	Normal	3	0.722	0.903	0.187	0.429	0.886	0.342
	Excessive		0.903	0.722	0.187	0.886	0.429	0.342
Growth form	Foliose	7	0.916	0.933	0.075	0.941	0.750	0.154
	Fruticose		0.933	0.916	0.075	0.750	0.941	0.154
Reproduction strategy	Apotecia	11	0.862	0.740	0.198	0.829	0.783	0.194
	Isidia		0.865	0.920	0.107	0.737	0.821	0.221
	Soredia		0.846	0.879	0.137	0.750	0.815	0.217
Phorophyte	Pinus	10	0.855	0.866	0.1398	0.677	0.889	0.216
	Quercus		0.866	0.855	0.1398	0.889	0.677	0.216
Bioclimatic belt	22a	8	0.911	0.902	0.093	0.721	0.933	0.172
	19c		0.902	0.911	0.093	0.933	0.721	0.172

Sn: sensitivity; Sp: specificity; Er: classification error; LVs: number of latent variables.

## Data Availability

Data are available at Appendix A.

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
