# Peer review of "Lichen Biodiversity and Near-Infrared Metabolomic Fingerprint as Diagnostic and Prognostic Complementary Tools for Biomonitoring: A Case Study in the Eastern Iberian Peninsula"

_jof, 2023, doi:10.3390/jof9111064_

Round 1

Reviewer 1 Report

Comments and Suggestions for Authors

Following the closure of the Andorra Thermal Power Plant in 2020 it was highly relevant to repeat the 1997 estimate of lichen response using the Index of Atmospheric  Purity (IAP), designed to detect changes in the lichen community in response to atmospheric SO2 deposition. The authors point out in the introduction that there are other changes in environmental conditions that have occurred since the 1997 survey. Data from the 6 air quality stations in the vicinity show changes in mean temperature and O3  concentrations at all air quality stations in the region.  The authors highlight changes in mean N deposition  and exceedance in Europe in S6,  but this data is not available in the stations despite there being major changes in Nitrogen deposition in response to changes in agricultural practices which are happening across Europe shown in S6.

The authors have shown changes in the response of the lichen communities over the 25 year period in IAP calculations (S5), ion selected species in the Damage Index (DI), and in total macrolichen diversity (S3). This data presents a puzzle as it demonstrates further deterioration at most sites except Corachar where species diversity is very low and there are few changes shown in lichen frequency in S3. There are interesting differences in the response on Pinus and Quercus where the decrease in 2022 of pollution indicators of Nitrogen including species of Xanthoria, Physcia and Hyperphyscia (species normally associated with high bark pH). You have looked at changes in the classification of species in relation to pH and to eutrophication (Figure 4 and S4) but these are not related to measured bark pH or N levels which may reveal another factor associated with the lichen response. There is an additional problem that is illustrated by the only crustose species recorded Lepra albescens which showed an increase on Quercus in 2022 at Toro where the IAP was highest in 1997. Crustose species are very slow growing and it would appear that this species has survived and increased in this site but that no other macrolichen species have increased. I would like to see bark pH data for 2022 as in regions where bark pH has been acidified by SO2 the exterior bark of Quercus is not lost and remains acid for a considerable period of time. This contrasts with the results on Pinus where older bark is regularly lost and both Usnea, Parmelia serrana and Pseudevernia species have shown an increase in abundance since 2020. Without bark pH data it is not possible to test this response!

The new methodology that has been evaluated out using NIR and NIRS to detect differences in categorical variables including growth forms, reproductive strategy, phorophyte and the bioclimatic belt is useful in that it shows the differential response of foliose and fruticose species and that geographical communities contribute to a different response across the area studied and could be used to provide an indication of pollution sensitivity of lichens.

You have provided mean annual estimates for SO2, O3 and temperature from the seven air quality monitoring stations shown in Figure 7.

However the lichen response results from 2022 need to be tested against actual levels of pollutants including NH3, NO2, and particulates and it appears that although particulates, NO2 and O3 are monitored across the region NH3 data is not available, and yet this is the pollutant most affecting rural areas.

Although the authors have collected lots of interesting data there remains the question of which factors are causing the changes in lichen response? The paper is well set out and the results indicate that there is a clear response to changing environmental conditions but these need to be tested against measured environmental conditions. If it is possible to further define these, they can be statistically tested in relation to both the IAP results, the species recorded and the categorical variables. The results would help to further define methods that can be used to detect changes in atmospheric and environmental conditions.  

Comments on the Quality of English Language

The English is good and only requires minor changes

Author Response

Referee 1

Following the closure of the Andorra Thermal Power Plant in 2020 it was highly relevant to repeat the 1997 estimate of lichen response using the Index of Atmospheric  Purity (IAP), designed to detect changes in the lichen community in response to atmospheric SO2 deposition. The authors point out in the introduction that there are other changes in environmental conditions that have occurred since the 1997 survey. Data from the 6 air quality stations in the vicinity show changes in mean temperature and O3 concentrations at all air quality stations in the region.  The authors highlight changes in mean N deposition and exceedance in Europe in S6, but this data is not available in the stations despite there being major changes in Nitrogen deposition in response to changes in agricultural practices, which are happening across Europe shown in S6.

Dear referee,

Thank you very much for your extensive and careful description of the work, we appreciate your comments and observations.

The data for nitrogen deposition at the air quality stations or nearby stations included in the ICP forest networks are indeed not available in the paper. The air quality stations included in the study record data for SO2, O3, temperature, wind speed, relative humidity and NO2. Only SO2, O3 and temperature data were included in the study, since the other parameters did not provide any additional information. It is true that nitrogen deposition is a key factor for the decline of lichen communities and that it would be very interesting to request these data from the ICP forest, but for the temporal analysis of the stations, as presented in the paper, we did not consider it necessary.

The authors have shown changes in the response of the lichen communities over the 25 year period in IAP calculations (S5), ion selected species in the Damage Index (DI), and in total macrolichen diversity (S3). This data presents a puzzle as it demonstrates further deterioration at most sites except Corachar where species diversity is very low and there are few changes shown in lichen frequency in S3. There are interesting differences in the response on Pinus and Quercus where the decrease in 2022 of pollution indicators of Nitrogen including species of Xanthoria, Physcia and Hyperphyscia (species normally associated with high bark pH). You have looked at changes in the classification of species in relation to pH and to eutrophication (Figure 4 and S4) but these are not related to measured bark pH or N levels which may reveal another factor associated with the lichen response. There is an additional problem that is illustrated by the only crustose species recorded Lepra albescens which showed an increase on Quercus in 2022 at Toro where the IAP was highest in 1997. Crustose species are very slow growing and it would appear that this species has survived and increased in this site but that no other macrolichen species have increased. I would like to see bark pH data for 2022 as in regions where bark pH has been acidified by SO2 the exterior bark of Quercus is not lost and remains acid for a considerable period of time. This contrasts with the results on Pinus where older bark is regularly lost and both Usnea, Parmelia serrana and Pseudevernia species have shown an increase in abundance since 2020. Without bark pH data it is not possible to test this response!

We agree with the reviewer's comments regarding the relevance of bark pH as a driver of possible changes in community diversity and composition. However, as this manuscript is more generally focused on investigating changes in epiphytic communities over time, such a detailed analysis of individual physico-chemical factors at the local tree scale is not a specific aim of the paper. We therefore prefer not to include this type of analysis in order to ensure a good balance of the topics covered: we believe that the paper is already complex enough in its current structure and that adding further details, while certainly worthy of consideration, would detract from the overall readability of the text. However, in order not to evade the reviewer's comment, we would like to point out that the variation in bark pH over a vast area and over a long period of time is also influenced by other factors that can sometimes be a source of discordant effects: in addition to the already mentioned acidification of rainfall, in the Mediterranean area the relationship between dry deposition (e.g. of basaltic dust), eutrophication resulting from local anthropic activities, the distribution of precipitation in stemflow vs. throughfall, etc. play an important role.

The new methodology that has been evaluated out using NIR and NIRS to detect differences in categorical variables including growth forms, reproductive strategy, phorophyte and the bioclimatic belt is useful in that it shows the differential response of foliose and fruticose species and that geographical communities contribute to a different response across the area studied and could be used to provide an indication of pollution sensitivity of lichens.

Regarding the NIRS methodology, although we have not been able to detect differences between healthy and damaged lichens, as the reviewer indicates, we have detected differential responses from foliose and fruticose species and in relation to the bioclimatic belts, which opens an area of study for the use of this technique as an indication of pollution sensitivity by lichens.

You have provided mean annual estimates for SO2, O3 and temperature from the seven air quality monitoring stations shown in Figure 7. However the lichen response results from 2022 need to be tested against actual levels of pollutants including NH3, NO2, and particulates and it appears that although particulates, NO2 and O3 are monitored across the region NH3 data is not available, and yet this is the pollutant most affecting rural areas.

Although it is true that NH3 data are very important to study impacts of pollution in rural areas, as I mentioned at the beginning of the answer. The air quality stations included in the study record SO2, O3, temperature, wind speed, relative humidity and NO2 data. Only SO2, O3 and temperature data were included in the study, as the other parameters did not provide any additional information.

Although the authors have collected lots of interesting data there remains the question of which factors are causing the changes in lichen response? The paper is well set out and the results indicate that there is a clear response to changing environmental conditions but these need to be tested against measured environmental conditions. If it is possible to further define these, they can be statistically tested in relation to both the IAP results, the species recorded and the categorical variables. The results would help to further define methods that can be used to detect changes in atmospheric and environmental conditions. 

On this point, the reviewer is probably right and we have tried to discuss these traits in the text. It must be said, however, that the stated aim of the work is not the detailed understanding of the individual factors that have determined the change in the diversity and composition of the epiphytic lichen communities in the study area, but rather the evaluation of their temporal variation “per se” (over the long term) using certain complementary approaches.

We therefore agree with the reviewer that biomonitoring methods would benefit from a more advanced understanding of the relative relationships between the effects of diversity drivers, but frankly we believe that answering this question requires “a priori” information that was not available, in our case, of a homogeneous manner across the area and over the long time period considered: from a conservative approach, without trying to oversell our results, we assume that lichen diversity is determined by a multifactorial complex of variables (which we will take into account using some more robust proxies), and we have focused on the temporal aspect of variation.

Reviewer 2 Report

Comments and Suggestions for Authors

Dear Authors,

I found your manuscript to be an enjoyable read. Overall, I have no major concerns, but I have some minor comments, most of which pertain to the figures. You will find these comments in the attached document.

Your manuscript is well-written, and it is important that the quality of the figures matches the overall standard. I kindly request that you dedicate some time to enhance the visual appeal of the figures, including those in the supplementary material. For example, Fig. S6 appears to be a screenshot due to the visible red line under 'throughfall.'

Additionally, I recommend rewriting all the figure legends. A good legend should convey the figure's content clearly, allowing readers to understand it without having to refer back to the manuscript. Please avoid using abbreviations in the figure legends since not all readers will recall them without referring to the text.

In terms of color choices, exercise caution. Opt for colors that reproduce well in print; for instance, consider filling the box plots in Fig. 2 instead of outlining them. In the case of PCoA plots for NIR data, the use of red and green might pose challenges for colorblind readers in distinguishing differences. It might be advisable to consider alternative color options.

I hope these suggestions will assist you in improving your manuscript.

Best regards,

Reviewer

Author Response

Referee 2

Dear Authors,

I found your manuscript to be an enjoyable read. Overall, I have no major concerns, but I have some minor comments, most of which pertain to the figures. You will find these comments in the attached document.

Your manuscript is well-written, and it is important that the quality of the figures matches the overall standard. I kindly request that you dedicate some time to enhance the visual appeal of the figures, including those in the supplementary material. For example, Fig. S6 appears to be a screenshot due to the visible red line under 'throughfall.'

Additionally, I recommend rewriting all the figure legends. A good legend should convey the figure's content clearly, allowing readers to understand it without having to refer back to the manuscript. Please avoid using abbreviations in the figure legends since not all readers will recall them without referring to the text.

In terms of color choices, exercise caution. Opt for colors that reproduce well in print; for instance, consider filling the box plots in Fig. 2 instead of outlining them. In the case of PCoA plots for NIR data, the use of red and green might pose challenges for colorblind readers in distinguishing differences. It might be advisable to consider alternative color options.

I hope these suggestions will assist you in improving your manuscript.

Best regards,

Reviewer

Dear referee, thank you so much for your comments and suggestions. We have tried to include all of them in the new manuscript (see ms_CC) and we have also replied your questions in the pdf you attached (see pdf_answered).

Regarding the Figures all of them have been modified, also the figure captions including the supplementary material. We hope that the modifications will be satisfactory to you.

In the case of PCoA plts for NIRS, we appreciate the reviewer’s concern for improving the inclusivity and readability of our paper. Besides colour, each point has a characteristic and unique shape. We have now completed the legends including the corresponding shape (see ms_CC).

Round 2

Reviewer 1 Report

Comments and Suggestions for Authors

The authors have improved the manuscript and responded to my queries by explaining that they do not have the data to answer these queries. I hope that this will lead to further investigation of the subject but agree that this paper is worth publishing as a good outline of an approach to the response of lichens to changing air quality.